# AA-SVD: Anchored and Adaptive SVD for Large Model Compression

## Abstract

Pretrained large-language and vision-language models have demonstrated remarkable capabilities over the years, but their ever-increasing size poses challenges for deployment and accessibility. Model compression offers a path toward democratizing access, yet many existing approaches either require costly retraining or result in substantial performance degradation. To address this, we introduce a fast SVD-based truncation framework for compressing pretrained networks that enables rapid compression of billion-parameter models without retraining. Unlike existing SVD-based approaches that optimize only on the original inputs — ignoring distribution shifts from upstream compression and thus propagating errors forward—or those that rely only on shifted inputs and risk drifting away from the original outputs, our approach accounts for both. By anchoring each compressed layer to the original outputs while explicitly modeling input distribution shifts, our method identifies optimal low-rank approximations that maintain functional equivalence with the uncompressed network, thereby preserving the behavior of the full model. Experiments across language and vision-language models of varying scales demonstrate that our method not only achieves favorable trade-offs between compression ratio and task accuracy, but also outperforms existing baselines particularly at low compression ratios—where the gap widens as compression becomes more aggressive—offering a practical solution for efficient, large-scale model deployment.

## 1 Introduction

The rapid progress of large-scale pretrained models has fundamentally transformed natural language processing and multimodal learning. Modern large language models (LLMs) (Touvron et al., 2023; Zhang et al., 2022; Achiam et al., 2023) and vision-language models (VLMs) (Radford et al., 2021; Liu et al., 2023; Dosovitskiy et al., 2021) now routinely contain billions of parameters, enabling strong generalization capabilities across a wide range of downstream tasks. However, this improvement in performance has come at the cost of scale: training, fine-tuning, and inference with such models often requires clusters of high-memory GPUs, making them prohibitively expensive to deploy in resource-constrained or latency-sensitive settings. As model sizes continue to grow, practical challenges around cost, efficiency, and accessibility become even more pressing Kaplan et al. (2020); Patterson et al. (2021).

One promising direction is to move beyond ever-larger models toward smaller, more efficient ones. Compact models can be trained from scratch for specialized tasks, but this approach sacrifices the broad generalization ability of large pretrained networks. Alternatively, smaller models can be obtained by *distilling* large networks into student models trained to mimic their behavior (Hinton et al., 2015; Xu et al., 2024), or by applying *post-training compression* techniques such as pruning, quantization, or low-rank factorization (Cheng et al., 2017; Zhu et al., 2024). While both approaches reduce memory footprint and inference cost, distillation typically requires substantial retraining data and compute (Jiao et al., 2020; Touvron et al., 2021), whereas post-training compression can often be applied more rapidly to pretrained networks (Frantar et al., 2022; Dettmers et al., 2022; Wang et al., 2025c), thereby offering a practical path towards democratizing deployment. These compressed models can either be deployed directly for efficient inference, fine-tuned to adapt to downstream tasks, or embedded within distributed systems that demand low-latency and high-throughput inference.

A wide range of model compression techniques have been proposed, spanning distinct methodological families. *Pruning* removes redundant weights or structures from neural networks, with early work on unstructured sparsification (Han et al., 2015) and the lottery ticket hypothesis (Frankle & Carbin, 2019) showing that smaller subnetworks can be retrained to match dense counterparts. While effective, pruning often requires iterative retraining and specialized sparsity-aware hardware to fully realize efficiency gains, though recent advances such as SparseGPT and its variants (Frantar & Alistarh, 2023; Ma et al., 2023; Ashkboos et al., 2024; An et al., 2024) have enabled post-training pruning of large language models. *Quantization* reduces numerical precision of weights and activations, thereby shrinking memory footprint and accelerating inference. Classic approaches demonstrated the feasibility of quantized neural networks (Hubara et al., 2017), while modern methods like LLM.int8() (Dettmers et al., 2022), QLoRA (Dettmers et al., 2023), and AWQ (Lin et al., 2024) allow near-lossless compression of transformers. However, quantization methods may require careful calibration and sometimes introduce instability for very low-bit settings. Another line of work leverages the inherent low-rank structure of network weights: *low-rank factorization* decomposes large matrices into compact representations, reducing both parameters and computation. Early applications in CNNs (Denton et al., 2014; Tai et al., 2015) demonstrated significant speedups, but naïve SVD truncation is known to degrade accuracy. More recent activation-aware approaches for LLMs (Yuan et al., 2023; Wang et al., 2025d; Li et al., 2025; Wang et al., 2025a; Li et al., 2025) explicitly account for input activations, mitigating this limitation at the cost of additional computation.

These methods differ in their retraining requirements, their dependence on large datasets versus small calibration samples, the efficiency with which compression can be applied to pretrained networks, the degree to which downstream accuracy is preserved, and the extent to which the resulting compressed structure aligns with modern accelerators (Cheng et al., 2018). Among these, *SVD-based methods* are especially appealing: they exploit the inherent low-rank structure of neural network weights, yielding compressed models without the need for expensive retraining (Denton et al., 2014; Jaderberg et al., 2014). A straightforward approach is to directly truncate weight matrices by retaining only the top singular components, but this often leads to severe degradation because it treats all input directions equally and discards information that is important for the actual distribution of activations (Denil et al., 2013; Chen et al., 2021; Wang et al., 2025d). This limitation has been repeatedly observed in large-scale networks, where naïve low-rank truncation fails to preserve task accuracy and generalization. To address this, activation-aware approaches have been developed that tailor the factorization to the input distribution, thereby retaining the directions most relevant to the network's operation. However, existing activation-aware SVD methods often optimize low-rank approximations using only the original input distribution (Yuan et al., 2023; Wang et al., 2025d; Li et al., 2025; Wang et al., 2025a), ignoring the shift introduced by upstream compression, which can propagate errors and degrade downstream performance. Conversely, methods that rely exclusively on shifted inputs, such as DobiSVD (Wang et al., 2025a), risk deviating from the original network behavior, introducing instability and loss of fidelity.

In this work, we present **AA-SVD**, a *fast SVD-based truncation framework* for compressing pretrained networks. Unlike existing SVD truncation or activation-aware methods that only consider a single input distribution, our approach accounts for both the original outputs and the distribution shifts caused by upstream compression. This design yields compressed layers that more faithfully preserve the functional behavior of the uncompressed model, enabling effective post-training compression of billion-parameter networks without retraining. Our contributions can be summarized as follows:

- **A fast compression method** that improves upon prior SVD-based approaches, with negligible overhead compared to optimization-heavy baselines such as DobiSVD Wang et al. (2025a).

- **A novel objective formulation** that anchors compressed layers to the original outputs while explicitly modeling input distribution shifts, thereby better preserving functional equivalence to the uncompressed model.

- **Comprehensive evaluation** across large-scale language models, demonstrating favorable trade-offs between compression ratio and accuracy, and outperforming existing SVD-based baselines.

## 2  RELATED WORK

Low-rank factorization, e.g., via singular value decomposition (SVD), has emerged as a promising direction for compressing large pretrained models. Compared to pruning or quantization, SVD-based methods offer several practical advantages. First, factorizing a weight matrix into low-rank components yields a *structured* representation that reduces both parameters and compute. The factorized form enables commuting multiplications— $(UV^\top)X = U(V^\top X)$ —which reduces memory requirement and can be implemented efficiently on existing accelerators. Second, unlike pruning, which often introduces irregular sparsity, or quantization, which requires specialized kernels for speedup, SVD-based methods produce dense but smaller matrices that integrate seamlessly with standard linear algebra libraries. Finally, they can be applied post-training with only small calibration samples (often a few hundred), making them particularly attractive for compressing billion-parameter models where retraining is infeasible. Recent methods such as ASVD (Yuan et al., 2023), SVD-LLM (Wang et al., 2025d), AdaSVD (Li et al., 2025), SVD-LLM V2 (Wang et al., 2025b) and Dobi-SVD (Wang et al., 2025a) have demonstrated the viability of this approach at scale in large language models.

Based on the optimization objective, SVD-based compression methods can be grouped into the following categories :

**Input-agnostic (direct) SVD.**  The simplest approach applies a truncated singular value decomposition to the weight matrix $W$, replacing it by a rank-$r$ approximation $W'$ constructed from its top singular components (Halko et al., 2011; Sainath et al., 2013). This method is appealing for its simplicity and minimal data dependence. However, direct SVD treats all input directions uniformly, ignoring the fact that in deep networks, the actual input activations $X$ lie in a highly anisotropic subspace. In such settings, the singular vectors preserved by SVD may not align with the task-relevant activation patterns or the dominant subspace of $X$, leading to suboptimal approximations. Indeed, empirical studies in neural network compression consistently find that direct SVD often underperforms data-aware variants tuned to activation statistics (e.g. Chen et al. (2021); Idelbayev & Carreira-Perpinán (2020)). More broadly, analyses of neural anisotropy directions suggest that deep models naturally concentrate representation into narrow subspaces, reinforcing why input-agnostic approximations are misaligned with the true geometry of activations Ortiz-Jiménez et al. (2020).

**Activation-aware factorization.**  To incorporate the geometry of the inputs actually seen by the network, activation-aware methods optimize the reconstruction

$$\min_{W':\mathrm{rank}(W')=r} \|WX - W'X\|_F^2,$$

where $X$ are activations collected from the original, uncompressed model. Examples include Drone (Chen et al., 2021), ASVD (Yuan et al., 2023), SVD-LLM (Wang et al., 2025d), AdaSVD (Li et al., 2025), and SVD-LLM V2 (Wang et al., 2025b). By preserving the action of $W$ on its occupied input subspace, these approaches are often more faithful than direct SVD. However, their performance hinges on the representativeness of the calibration set used to obtain $X$. If calibration data are narrow or unaligned with downstream usage, compressed models may overfit to the sampled geometry and fail to generalize. Related activation-matching objectives also appear in structured pruning frameworks, such as FLAP (An et al., 2024), which similarly leverage activation statistics to guide parameter removal.

**Shift-aware factorization.**  A key limitation of activation-aware approaches is that they optimize with respect to the original activations $X$, even though, in a sequentially compressed model, later layers actually receive shifted inputs $X'$. To account for this, shift-aware methods, e.g. Dobi-SVD (Wang et al., 2025a), optimize

$$\min_{W':\mathrm{rank}(W')=r} \|WX' - W'X'\|_F^2,$$

using activations from the partially compressed network. By aligning the approximation to the distribution the layer truly encounters, these methods can mitigate error propagation through the stack. Their drawback, however, is that when upstream compression has already degraded representations, anchoring solely to $X'$ risks amplifying divergence from the original mapping. In addition, batch-based surrogates for $X'$ are often noisy or unrepresentative, which can introduce instability into the

approximation. As a result, shift-aware objectives alone provide only a partial solution. Related ideas also appear implicitly in earlier CNN low-rank factorization (Denton et al., 2014; Jaderberg et al., 2014), where activations were collected after partial compression, and in layer-wise distillation methods (e.g., TinyBERT (Jiao et al., 2020)), where the compressed model is aligned to the teacher using its own inputs.

Beyond the choice of approximation objective, the effectiveness of low-rank factorization depends critically on how ranks are distributed across layers. Uniform allocation ignores heterogeneity in both compressibility and functional importance. Adaptive strategies such as AdaSVD (Li et al., 2025) leverage layer-importance signals to allocate more rank where needed, in line with importance-based pruning approaches such as ShortGPT (Men et al., 2024). SVD-LLM V2 (Wang et al., 2025b) instead proposed a heuristic that reallocates rank based on the truncation loss $\|WX - W'X\|_F^2$ observed after uniform compression. Earlier work on CNNs has also explored learning per-layer ranks directly via group sparsity regularization over singular values (Idelbayev & Carreira-Perpiñán, 2020), showing clear gains over uniform allocation. Differentiable allocation schemes have also been explored (e.g., in Dobi-SVD (Wang et al., 2025a)), but these typically require costly optimization and rely on unstable batch-level statistics. Collectively, these advances highlight that compression quality depends not only on the local objective but also on *where* and *how* rank is assigned.

## 3 AA-SVD

In this section we present our compression framework, **AA-SVD** (Anchored and Adaptive SVD). The central idea is to construct low-rank approximations of each linear transformation in a pretrained network such that the compressed model remains *locally faithful* to the original network, while simultaneously adapting to the distributional shifts induced by upstream compression.

Formally, we denote a weight matrix at layer $\ell$ by $W \in \mathbb{R}^{m \times n}$, with input activations $X \in \mathbb{R}^{n \times k}$ and outputs $WX \in \mathbb{R}^{m \times k}$, where $k$ is the number of calibration samples. After compressing earlier layers, the same layer instead receives shifted activations $X' \in \mathbb{R}^{n \times k}$, producing outputs $W'X'$. Our objective is to replace $W$ with a rank-constrained approximation $W' \in \mathbb{R}^{m \times n}$, where $\mathrm{rank}(W') = r \ll \min(m, n)$, such that $W'X'$ remains close to $WX$. In this way, **AA-SVD** enforces that the compressed layer continues to behave like the original one *in the local neighborhood defined by its actual inputs*, while still anchored to the outputs of the uncompressed model.

### 3.1 OBJECTIVE

Our goal is to compress each linear transformation while ensuring that the resulting network remains *locally faithful* to the original model under the inputs it will actually encounter. Concretely, for a weight matrix $W \in \mathbb{R}^{m \times n}$ with original inputs $X \in \mathbb{R}^{n \times k}$ and shifted inputs $X' \in \mathbb{R}^{n \times k}$ (after upstream compression), we seek a low-rank approximation $W' \in \mathbb{R}^{m \times n}$ that solves

$$\min_{W':\mathrm{rank}(W')=r} \|WX - W'X'\|_F^2.$$

This objective enforces that the compressed outputs $W'X'$ stay close to the original outputs $WX$, anchoring the compressed network to the behavior of the uncompressed one while simultaneously adapting to the shifted input distribution. By explicitly constraining $\mathrm{rank}(W') = r$, the problem is well-posed as a low-rank regression: we seek the best rank–$r$ approximation of the mapping from $X'$ to $WX$.

**Theorem 3.1** (Low-rank approximation with upstream-modified inputs). *Let $W \in \mathbb{R}^{m \times d}$ be a fixed weight matrix and $X, X' \in \mathbb{R}^{d \times N}$ be two sets of input activations (columns are samples). Define*

$$A := XX'^\top \in \mathbb{R}^{d \times d}, \qquad B := X'X'^\top \in \mathbb{R}^{d \times d}.$$

*Fix a target rank $k \in \mathbb{N}$. Consider the optimization problem*

$$\min_{\mathrm{rank}(W') \leq k} \left\| WX - W'X' \right\|_F^2. \tag{1}$$

*Let $B = R^\top R$ be a Cholesky factorization with $R$ upper triangular, and define $M := WAR^{-1}$. If $M = U\Sigma V^\top$ is a thin singular value decomposition, then an optimal solution to equation 1 is*

$$W'^\star = \left(U_k \Sigma_k V_k^\top\right) R^{-1},$$

---

**Algorithm 1 AA-SVD** Low-rank compression

---

**Require:** Weight matrix $W \in \mathbb{R}^{m \times d}$, original inputs $X \in \mathbb{R}^{d \times N}$, current inputs $X' \in \mathbb{R}^{d \times N}$, target rank $k$
1: Compute covariances $A = XX'^{\top}$ and $B = X'X'^{\top}$
2: Cholesky factorization: $B = R^{\top}R$
3: Compute $M = WAR^{-1}$
4: Truncated SVD: $M \approx U_k \Sigma_k V_k^{\top}$
5: Return $W' = (U_k \Sigma_k V_k^{\top})R^{-1}$ or factorized matrices $U = U_k \Sigma_k$ and $V = V_k^{\top}R^{-1}$

---

*where $U_k, \Sigma_k, V_k$ are the top-$k$ blocks of the SVD. The minimum objective value is*

$$\|WX\|_F^2 - \|M\|_F^2 \; + \sum_{i>k} \sigma_i(M)^2,$$

*where $\sigma_i(M)$ are the singular values of $M$.*

*Proof.* Expanding the squared Frobenius norm gives

$$\|WX - W'X'\|_F^2 = \mathrm{tr}(W'BW'^{\top}) - 2\,\mathrm{tr}(WAW'^{\top}) + \|WX\|_F^2.$$

Since $B = R^{\top}R$, the first term is $\|W'R\|_F^2$. Completing the square yields

$$\|W'R - WAR^{-1}\|_F^2 - \|WAR^{-1}\|_F^2 + \|WX\|_F^2.$$

Thus minimizing equation 1 is equivalent to minimizing $\|W'R - M\|_F^2$ subject to $\mathrm{rank}(W'R) \leq k$, where $M = WAR^{-1}$. Because $R$ is invertible, $\mathrm{rank}(W'R) = \mathrm{rank}(W')$. By the Eckart–Young–Mirsky theorem, the optimal approximation is $U_k \Sigma_k V_k^{\top}$, yielding

$$W'^{\star} = (U_k \Sigma_k V_k^{\top})R^{-1},$$

and the minimal value as claimed. $\qquad\square$

**Corollary 3.2** (Classical whitening as a special case). *If $X' = X$, then $A = B$ and $M = WB^{1/2} = WR^{\top}$. The solution reduces to*
$$W'^{\star} = (WB^{1/2})_k \, B^{-1/2},$$
*the standard whitening-based low-rank regression solution.*

*Remark* 3.3 (Rank-deficient $X'$). If $B \succeq 0$ is singular, the Cholesky factorization does not exist. In this case replace $R^{-1}$ by the Moore–Penrose factor $B^{+1/2}$, or equivalently use a Tikhonov-regularized factorization $B + \varepsilon I = R_\varepsilon^{\top} R_\varepsilon$ and let $\varepsilon \to 0^+$. The same argument then shows that

$$W'^{\star} = \left(U_k \Sigma_k V_k^{\top}\right) B^{+1/2}, \qquad M := WAB^{+1/2},$$

is a minimum-norm optimizer, with minimal value given by the same formula.

Theorem 3.1 establishes that the optimal rank-$k$ compressed operator is obtained by whitening the modified inputs $X'$ via their covariance, projecting the cross-term $WA$ into this whitened space, applying truncated SVD, and mapping back. This closed-form solution generalizes the classical whitening construction ($X' = X$) and can be implemented efficiently with a Cholesky factorization. Importantly, our formulation operates only on the covariance matrices $XX'^{\top}$ and $X'X'^{\top}$ rather than the raw activations themselves. This is especially advantageous when the number of samples is large (e.g. in our setting with 256 samples of length 2048, corresponding to over half a million effective columns), since the covariance matrices are fixed-size $d \times d$ regardless of the batch length. For clarity, Algorithm 1 summarizes the procedure.

## 4 EXPERIMENTS

We empirically evaluate our method on large-scale language models from the LLaMA family, focusing primarily on LLaMA-7B and extending to larger variants to assess scalability. Our goals are threefold: (i) to compare against existing SVD-based and low-rank baselines in terms of perplexity

Table 1: Comparison of **AA-SVD** with SOTA methods for SVD-based compression of Llama-7B on two language modeling tasks and six common sense reasoning datasets (zero-shot evaluation). Best performance is marked in bold. [†] uses LoRA fine-tuning, while [‡] uses dynamic or non-uniform ratio allocation.

| Ratio | Method | PPL ($\downarrow$) | | Accuracy ($\uparrow$) | | | | | |
|---|---|---|---|---|---|---|---|---|---|
| | | Wiki2 | PTB | Openb. | ARC_e | ARC_c | WinoG. | PIQA | MathQA |
| 1.0 | Baseline | 5.68 | 8.79 | 0.34 | 0.75 | 0.42 | 0.69 | 0.79 | 0.27 |
| 0.8 | ASVD | 11.14 | 16.55 | 0.25 | 0.53 | 0.27 | 0.64 | 0.68 | 0.24 |
| | SVD-LLM[†] | 7.94 | 16.22 | 0.22 | 0.58 | 0.29 | 0.63 | 0.69 | 0.24 |
| | Dobi-SVD[‡] | 8.54 | **14.83** | 0.26 | 0.59 | 0.31 | **0.66** | **0.70** | 0.23 |
| | AA-SVD | **7.67** | 16.11 | **0.29** | **0.64** | **0.33** | 0.65 | 0.69 | **0.24** |
| 0.6 | ASVD | 1407 | 3292 | 0.13 | 0.28 | 0.22 | 0.48 | 0.55 | 0.19 |
| | SVD-LLM[†] | 13.11 | 63.75 | 0.19 | 0.42 | 0.25 | 0.58 | 0.60 | 0.21 |
| | Dobi-SVD[‡] | 13.54 | 46.38 | **0.22** | 0.41 | **0.27** | 0.58 | **0.61** | 0.23 |
| | AA-SVD | **12.19** | **35.32** | 0.19 | **0.46** | 0.23 | **0.59** | 0.60 | **0.23** |
| 0.4 | ASVD | 57057 | 45218 | 0.12 | 0.26 | 0.21 | 0.49 | 0.53 | 0.18 |
| | SVD-LLM[†] | 53.74 | 438.58 | 0.14 | 0.28 | **0.22** | 0.50 | **0.55** | 0.21 |
| | Dobi-SVD[‡] | 46.18 | 238.91 | 0.15 | 0.31 | 0.20 | **0.52** | 0.54 | 0.22 |
| | AA-SVD | **29.54** | **214.84** | 0.15 | **0.32** | 0.20 | 0.50 | 0.54 | **0.22** |
| 0.2 | SVD-LLM[†] | 1349 | – | 0.07 | 0.03 | – | 0.04 | 0.07 | 0.01 |
| | AA-SVD | **144.03** | **394.52** | **0.14** | **0.28** | **0.22** | **0.51** | **0.52** | **0.22** |

and downstream reasoning accuracy, (ii) to quantify efficiency improvements in memory footprint and inference cost, and (iii) to analyze the contribution of different design choices, including calibration set size, dynamic rank allocation, and post-compression refinements. Unless noted otherwise, all compression methods use a calibration set of 256 samples drawn from the WikiText2 dataset, following prior work. Performance is evaluated using two complementary metrics: (i) *language modeling perplexity*, measured on standard corpora including WikiText2 (Merity et al., 2016), and PTB (Marcinkiewicz, 1994); and (ii) *accuracy on commonsense reasoning*, measured on benchmarks such as Winogrande (Sakaguchi et al., 2020), PIQA (Bisk et al., 2020), MathQA (Amini et al., 2019), ARC-Easy and ARC-Challenge (Clark et al., 2018), and OpenBookQA (Mihaylov et al., 2018).

## 4.1 MAIN RESULTS

We evaluate the performance of **AA-SVD** with compression ratios ranging from 20% to 80%. Table 1 reports perplexity on two language modeling corpora (WikiText2 and PTB) and accuracy across six common sense reasoning benchmarks, under varying compression ratios. We compare against other SVD-based compression methods - ASVD, SVD-LLM, and DoBi-SVD.

At a high compression ratio of 0.8, AA-SVD already improves over all baselines in terms of average accuracy while maintaining perplexity close to the best-performing methods. For instance, AA-SVD yields the lowest perplexity of 7.67 and higher reasoning accuracy than DoBi-SVD on four out of six tasks, demonstrating robustness across both metrics.

As compression becomes more aggressive, the gap between AA-SVD and competing methods widens. At ratio 0.6, AA-SVD reduces perplexity substantially (WikiText2: 12.19 vs. 13.54 for DoBi-SVD, while PTB: 35.32 vs. 46.38), while either matching or outperforming in reasoning accuracy. At ratio 0.4, AA-SVD achieves a perplexity reduction of nearly 20% over DoBi-SVD and consistently ranks among the top two methods on all reasoning tasks.

The advantage is most pronounced at the extreme ratio of 0.2. Here, competing approaches collapse, with SVD-LLM reporting almost degenerate results. In contrast, AA-SVD remains functional, preserving non-trivial accuracy (e.g., PIQA: 0.51, ARC_c: 0.22) and maintaining perplexities below

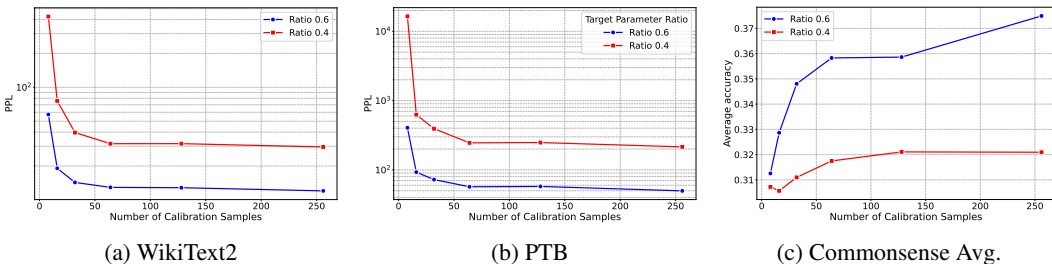

| (a) WikiText2 | (b) PTB | (c) Commonsense Avg. |

Figure 1: Impact of calibration set size on compression performance. Performance is measured by perplexity on WikiText2 and PTB, and average accuracy across six commonsense reasoning tasks.

400. This indicates that the combination of shift anchoring and dynamic rank allocation stabilizes compression even in highly resource-constrained regimes.

Overall, these results highlight that AA-SVD offers the best trade-off between language modeling fidelity and downstream reasoning ability, especially when compression is aggressive. Notably, our method avoids the collapse observed in prior SVD-based approaches at low ranks, underscoring the importance of accounting for both original outputs and shifted inputs.

## 4.2 MEMORY AND SPEEDUP

Low-rank factorization reduces both parameter count and compute cost by replacing a dense matrix with the product of two thin factors. Consider a linear layer $W \in \mathbb{R}^{m \times n}$. The original layer requires $mn$ parameters and $O(mn)$ FLOPs per forward pass. A rank-$r$ factorization stores $mr + nr$ parameters and incurs $O(mr + nr)$ FLOPs, which is cheaper whenever $r \ll \min(m, n)$. The effective compression ratio is

$$\rho = \frac{mr + nr}{mn}.$$

For example, with $m = n = 4096$ and $r = 512$ ($\rho = 0.125$), the parameter count drops from 16.8M to 4.2M (a $4\times$ reduction), and FLOPs per forward pass reduce by the same factor.

Beyond weights and FLOPs, low-rank factorization can also reduce the memory footprint of the key–value (KV) cache during autoregressive inference. Since attention projections are compressed, the activations stored in the cache scale with $r$ rather than $n$, yielding proportional savings in both memory and bandwidth. As highlighted in SVD-LLM (Wang et al., 2025d) and follow-up works, this reduction is crucial for long-context inference where KV-cache dominates memory usage.

Our method (AA-SVD) preserves this structural efficiency: the cost of computing compressed weights is incurred once during compression, while inference cost and KV-cache size match those of standard low-rank layers. Thus, AA-SVD offers the same runtime and memory benefits as prior SVD-based methods, with its main advantage lying in improved approximation quality under aggressive compression.

## 4.3 ABLATIONS AND ANALYSIS

**Impact of Number of Calibration Samples.** Figure 1 illustrates the impact of calibration set size on compression performance. We report perplexity on WikiText2 (Fig. 1a) and PTB (Fig. 1b), as well as the average accuracy across six reasoning tasks (Fig. 1c), for compression ratios 0.6 and 0.4. Performance improves steadily with additional samples, but perplexity quickly saturates beyond ~64 examples. Notably, even with as few as 64 samples, AA-SVD remains stable and delivers competitive results, indicating that only a modest calibration set is required. For commonsense reasoning, particularly at the higher compression ratio, larger calibration sets provide incremental gains, suggesting room for further improvement in more data-rich settings.

**Error Evolution Across Layers.** To better understand how compression affects the internal representations, we track the discrepancy between the original and compressed models across depth. Figure 2 plots layerwise *cosine distance* between original and compressed features ($WX$ vs. $W'X'$;

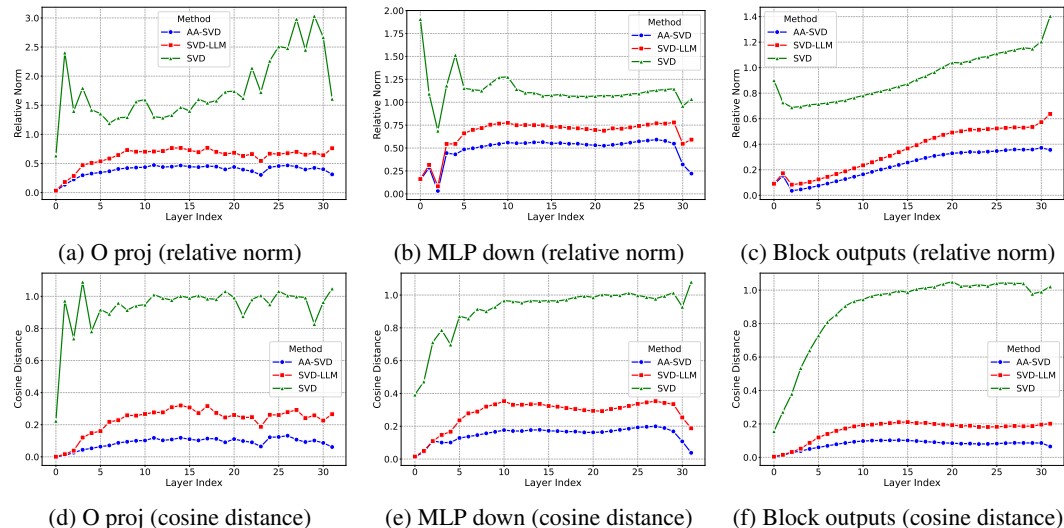

(a) O proj (relative norm)    (b) MLP down (relative norm)    (c) Block outputs (relative norm)

(d) O proj (cosine distance)    (e) MLP down (cosine distance)    (f) Block outputs (cosine distance)

Figure 2: Layerwise error evolution. Top row: relative norm difference $\|WX - W'X'\|_F / \|WX\|_F$. Bottom row: cosine distance between $WX$ and $W'X'$. Results are shown separately for query (Q) projections, MLP down-projections, and block outputs.

lower is better) alongside the *relative norm* error $\|WX - W'X'\|_F / \|WX\|_F$. We compress Llama-7B model at $60\%$ compression ratio with **AA-SVD** and compare it with naive SVD as well as SVD-LLM. We use only $64$ calibration samples from WikiText2 for each method. We show results for output projections, MLP down-projections, and transformer layer/block outputs. Across all methods, AA-SVD consistently achieves the **lowest cosine distance** and **lowest relative norm** error, while direct SVD exhibits the largest divergences, especially in deeper layers where error accumulates. SVD-LLM lies in between but still shows increasing gap with depth. This reduction in layerwise error directly translates into stronger end-task performance. For example, AA-SVD achieves a perplexity of $12.92$ on WikiText2 and $57.02$ on PTB, compared to ($14.38$ / $77.71$) for SVD-LLM and ($50714$ / $60103$) with naive SVD (indicating catastrophic degradation). These results confirm that stabilizing error growth across depth is critical for preserving downstream accuracy. Anchoring to both original outputs and shifted inputs curbs error growth and preserves feature geometry throughout the network.

## 5 CONCLUSION

We introduced a fast, post-training framework for compressing large language and vision-language models using rank-constrained SVD. Unlike prior approaches that rely exclusively on original inputs or shifted activations, our method unifies both perspectives: it anchors each compressed layer to the outputs of the uncompressed network while adapting to the inputs that arise after upstream compression. This leads to closed-form solutions with a rank constraint, efficient to compute from a small calibration set. Extensive experiments on the LLaMA family and commonsense reasoning benchmarks show that our approach consistently outperforms direct and activation-aware SVD methods, as well as shift-only approaches such as DobiSVD. At low compression ratios, our method preserves accuracy with negligible loss, while under aggressive compression it widens the gap to baselines. Overall, our study demonstrates that careful design of the compression objective and rank allocation strategy enables billion-parameter models to be compressed quickly and effectively without retraining. We hope this work contributes toward practical, accessible deployment of large-scale pretrained models, and inspires further exploration of hybrid objectives and allocation schemes for efficient model compression.

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
