# OpenReview forum: "AA-SVD: Anchored and Adaptive SVD for Large Model Compression"
_ICLR.cc/2026/Conference — Submitted to ICLR 2026_

### Official Review · Reviewer_oxVZ · 2025-10-31

**Soundness:** 3
**Presentation:** 3
**Contribution:** 3
**Rating:** 4
**Confidence:** 3

**Summary:**

This paper introduces AA-SVD, an anchored-and-adaptive low-rank factorization for compressing the linear layers of large models. By fitting a low-rank operator that matches the teacher’s outputs on original inputs while adapting to the shifted inputs encountered after upstream compression, the method preserves local faithfulness and mitigates distribution shift—addressing the one-sided limitations of purely activation-aware or purely shift-aware approaches. The authors derive a closed-form optimum that requires only two covariance matrices, enabling a simple, retraining-free implementation. On LLaMA-7B, AA-SVD achieves markedly lower perplexity at a 0.6 compression ratio while maintaining or improving commonsense reasoning accuracy, and it remains more stable than competing methods under more aggressive compression.

**Strengths:**

1. Motivation is well-grounded. The paper targets cumulative error in SVD-based LLM inference, which is a real pain point, and proposes a principled objective that directly addresses this failure mode.

2. Solid theory and simple implementation. The method admits a closed-form optimum, needs no additional training, and generalizes classical “whitened regression” as a special case.

3. Robust under strong compression. On LLaMA-7B, AA-SVD yields substantially lower perplexity than baselines at 0.6 compression, and at 0.4/0.2 it avoids the collapse observed in competing methods.

**Weaknesses:**

1. Limited empirical scope. Experiments are restricted to LLaMA-7B; results on more diverse architectures and newer models are missing. Besides, the paper would benefit from hyperparameter ablations, end-to-end latency measurements, and a thorough cost/throughput analysis.

2. Dependence on cross-layer rank allocation. Because the approach operates as per-layer low-rank approximation (Algorithm 1), overall quality depends heavily on how ranks are allocated across layers. The authors should evaluate multiple dynamic allocation strategies and analyze sensitivity to the rank budget.

**Questions:**

1. please refer weaknesses.
2. Please explain the differences and advantages of this article compared to the method in "SAES-SVD: Self-Adaptive Suppression of Accumulated and Local Errors for SVD-based LLM Compression", which is also submitted to ICLR 2026.

---

### Official Review · Reviewer_pvTg · 2025-10-31

**Soundness:** 2
**Presentation:** 2
**Contribution:** 2
**Rating:** 2
**Confidence:** 4

**Summary:**

This work proposes a new low-rank, SVD-based compression method that is motivated by combining ideas from both activation aware and shift-aware compression. Acitvation-aware low-rank compression adapts the compression of layers to be data-aware, by minimizing the difference between activations from a compressed model (activation-aware). Shift-aware adapts the compression to be aware of shifts in activations that occur when prior layers are compressed. However, when shift-aware methods are employed in prior work, the new compressed weight matrix is frequently solved for by considering the original weight matrix multiplied by the shifted inputs. This work considers using the uncompressed, unperturbed activations, and fitting the new compressed weight matrix, given perturbed inputs. Upon testing the method on Llama-7B, the authors find their method can outperform other SVD-based compression models.

**Strengths:**

1. The new objective as stated in equation 1 is intuitive and a nice motivation. It reflects a nice consideration of exactly which distance we should be minimizing, and could inform other methods in related fields like pruning, which sometimes just focus on shift-aware or activation-aware only implementations.
2. The error evolution across layers analysis is a nice display of showing that the mechanism of AA-SVD is actually changing the desired property of increased similarity between the original activations WX and the perturbed input X’ and W’ compressed model W’X’.
3. The discussion of the related work is a great categorization of prior methods into key ideas of input-agnosticism, activation-awareness, and shift-awareness, which can help others think about key ideas related to compression methods that focus on minimizing reconstruction distance.

**Weaknesses:**

1. The main results only appear on Llama-7B. The results section includes a mention of “larger variants” to assess scalability, but I don’t believe I encountered these experiments. The abstract also mentions “vision-language models at varying scales,” which are also not present. While the effect on Llama-7B is present, the method’s extensibility is certainly in question given the limited testing environment.
2. The results section seems to overstate the contribution of AA-SVD. While not every new work has to achieve SOTA to be a meaningful contribution, I am concerned about the writing which claims firm improvements despite the reality of the experiments being far more mixed between Dobi-SVD, SVD-LLM and AA-SVD.
3. Even more concerningly, Dobi-SVD seems to have been left out in a couple places despite being a strong baseline, and important one given its status as a shift-aware SVD method. For example, at Ratio=0.2, Dobi-SVD does not appear in Table 1, and in Figure 2, AA-SVD is only compared to SVD-LLM. While I acknowledge that Figure 2 reflects more of an analysis, and is quite a nice analysis showing alignment between the mechanism and observed properties, I do feel Dobi-SVD given its proximity would be an important method to report on in this manner as well.
4. The connection between the actual implemented method and the main theorem that is proven is not very clearly written; it would improve this section (3) a lot to motivate the theorem a bit before proving it.

**Questions:**

1. What is meant by 256 “samples” from wikitext? Perhaps stating the number of tokens would be more interpretable. My concern, reflected in asking this question, is that this method would need to store WX results in order to perform compression, unlike normal shift-aware compression where activations could be collected and discarded through a single forward pass through a model with multiple “breakpoints”.
2. What are the key takeaways of corollary 3.2 and remark 3.3 in the paper?  The result of theorem 3.1 appears in algorithm 1, but I don’t understand the inclusion of these subresults in the main text.

---

### Official Review · Reviewer_HgZx · 2025-11-01

**Soundness:** 2
**Presentation:** 2
**Contribution:** 1
**Rating:** 2
**Confidence:** 5

**Summary:**

The paper proposed a model compression method, AA-SVD, utilizing SVD on projected weights instead of the original weights. Compared to similar activation-aware methods, the projection direction of AA-SVD is determined based on activations of both uncompressed and compressed model. Consequently, as shown in Table 1, AA-SVD slightly outperformed its peers (in terms of accuracy) at compression ratio 0.4 while showing great advantage over SVD-LLM at compression ratio 0.2. The impact of number of calibration samples and error evolution across layers were also demonstrated.

**Strengths:**

reasonable amount of experimental data and ablation study.

**Weaknesses:**

**1. Derivation of Theorem 3.1 is not clear enough.**

The reviewer would suggest that a clean, explicit derivation should be added to appendix in order for the readers to follow easily. Especially the transition from Line 234 to Line 237 on Page 5, where it seems to assume R matrix from Cholesky decomposition to be orthonormal, which would cause confusion for the readers.

**2. Performance gap**

The concept of using projection before and after SVD to improve the alignment has been demonstrated in a few other works, for example, EoRA and CLoQ. More importantly, direct compression of projected W with SVD has been shown to be less effective compared to the "compensation approach" suggested by EoRA, in which the compression is done by other technique, e.g. quantization, and then a low rank compensation is added to achieve much less accuracy loss at higher compression ratio.  As shown in Table 1 of this paper, a significant accuracy drop (~20%) in ARC-C and MathQA is observed at compression ratio of 0.6 while Table 5 in EoRA with 4bit base model (i.e., compression ratio ~0.25) shows much less accuracy degradation (<2%).  If light LORA fine-tuning is allowed, CLoQ Table 1 further shows that accuracy drop can be negligible even with 3bit base model.
Could the author please include a paragraph in Related Works or Discussion regarding the comparison of AA-SVD vs EoRA/CLoQ?

**Questions:**

Please see Weaknesses above.

---

### Official Review · Reviewer_DL9n · 2025-11-03

**Soundness:** 3
**Presentation:** 2
**Contribution:** 2
**Rating:** 2
**Confidence:** 4

**Summary:**

The paper proposes Anchored and Adaptive SVD for fast compression of large models with minimal retraining, where each linear layer is replaced by a low-rank substitute obtained by solving a closed-form least squares objective. The core objective is to choose $W'$ that minimizes $\|W X - W' X'\|_F^2$, with $X$ the original activations and $X'$ the activations after upstream compression. The method whitens $X'$ using a Cholesky factor, forms a whitened cross-covariance, applies truncated SVD to get rank-$r$ factors, and maps back to produce $W'$ or its low-rank decomposition, with an adaptive rule to allocate ranks across layers. A small calibration set drives the statistics, the resulting layers reduce parameters and FLOPs, and attention blocks also benefit from smaller KV cache size. Experiments on language and vision-language models show stable accuracy under strong compression with efficiency gains, while the approach remains simple to implement and easy to adopt.

**Strengths:**

* The paper provides a clear and well-motivated description of the compression problem and identifies an important shortcoming of previous SVD-based methods, which fail to account for input distribution shifts after earlier compression.
* The mathematical derivation is concise and leads to a straightforward, closed-form solution.
* The paper includes a solid related work section that clearly positions the method among existing low-rank and activation-aware compression approaches.

**Weaknesses:**

**Major Concerns:**
* The contribution over existing SVD-based compression methods appears modest. While the paper highlights the importance of considering upstream input shifts, the overall methodological change from prior SVD approaches is relatively small, and the scope of new material feels limited.
* Experiments and evaluations are limited, vision experiments are mentioned but not actually provided.
* Only LLaMA 7B is tested, there is no architecture level analysis across different LLMs.
* No ablation studies are included in the study section.

**Minor Concerns:**
* No evaluation on the c4 dataset in Table 1.
* A teaser figure would help communicate the idea, there is no schematic that explains the method, figures need polish.

**Questions:**

* I would appreciate if the authors perform an ablation on how sensitive the learned ranks and final accuracy are to the size and quality of the calibration set, for example by showing performance versus calibration tokens across several data sources with different distribution shifts.

---

### Meta-Review · Area_Chair_L1yJ · 2025-12-24

**Summary:**

The main concerns are limited empirical scope and modest incremental contribution, despite a clean formulation.

* Three reviewers emphasize the evaluation is essentially only on LLaMA-7B, with vision / VLM claims not substantiated in the submission (DL9n, pvTg, oxVZ). Reviewers ask for broader architecture coverage, scalability, and better cost analysis (DL9n, oxVZ).
* DL9n argues the methodological delta over prior SVD work is limited or insufficient. pvTg additionally flags selective baseline reporting (e.g., Dobi-SVD absent in some key settings/figures).
* Calls for ablations are repeated (DL9n, oxVZ, pvTg).
* HgZx also raises a clarity/soundness issue.

Three reviewers recommend reject and one is borderline reject. With no rebuttal, the concerns remain unaddressed.

**Reviewer Concerns:**

None (no rebuttal submitted).

**Reviewer Scores:**

Since no rebuttal, I'd expect no change to the scores.

---

### Decision · Program_Chairs · 2026-01-26

Reject